# Low-Level Endothelial TRAIL-Receptor Expression Obstructs the CNS-Delivery of Angiopep-2 Functionalised TRAIL-Receptor Agonists for the Treatment of Glioblastoma

**DOI:** 10.3390/molecules26247582

**Published:** 2021-12-14

**Authors:** Nivetha Krishna Moorthy, Oliver Seifert, Stephan Eisler, Sara Weirich, Roland E. Kontermann, Markus Rehm, Gavin Fullstone

**Affiliations:** 1Institute of Cell Biology and Immunology, University of Stuttgart, Allmandring 31, 70569 Stuttgart, Germany; nivetha.k@izi.uni-stuttgart.de (N.K.M.); oliver.seifert@izi.uni-stuttgart.de (O.S.); stephan.eisler@srcsb.uni-stuttgart.de (S.E.); roland.kontermann@izi.uni-stuttgart.de (R.E.K.); markus.morrison@izi.uni-stuttgart.de (M.R.); 2Stuttgart Research Center Systems Biology, University of Stuttgart, Nobelstrasse 15, 70569 Stuttgart, Germany; 3Institute of Biochemistry and Technical Biochemistry, University of Stuttgart, Allmandring 31, 70569 Stuttgart, Germany; sara.weirich@ibtb.uni-stuttgart.de

**Keywords:** glioblastoma, TRAIL, angiopep-2, CNS delivery, receptor-mediated transcytosis, blood–brain barrier

## Abstract

Glioblastoma (GBM) is the most malignant and aggressive form of glioma and is associated with a poor survival rate. Latest generation Tumour Necrosis Factor Related Apoptosis-Inducing Ligand (TRAIL)-based therapeutics potently induce apoptosis in cancer cells, including GBM cells, by binding to death receptors. However, the blood–brain barrier (BBB) is a major obstacle for these biologics to enter the central nervous system (CNS). We therefore investigated if antibody-based fusion proteins that combine hexavalent TRAIL and angiopep-2 (ANG2) moieties can be developed, with ANG2 promoting receptor-mediated transcytosis (RMT) across the BBB. We demonstrate that these fusion proteins retain the potent apoptosis induction of hexavalent TRAIL-receptor agonists. Importantly, blood–brain barrier cells instead remained highly resistant to this fusion protein. Binding studies indicated that ANG2 is active in these constructs but that TRAIL-ANG2 fusion proteins bind preferentially to BBB endothelial cells via the TRAIL moiety. Consequently, transport studies indicated that TRAIL-ANG2 fusion proteins can, in principle, be shuttled across BBB endothelial cells, but that low TRAIL receptor expression on BBB endothelial cells interferes with efficient transport. Our work therefore demonstrates that TRAIL-ANG2 fusion proteins remain highly potent in inducing apoptosis, but that therapeutic avenues will require combinatorial strategies, such as TRAIL-R masking, to achieve effective CNS transport.

## 1. Introduction

Glioblastoma (GBM), a grade IV astrocytoma [1], is the most prevalent and deadly form of primary cancer of the central nervous system (CNS) with the rate of incidence of 3.22 cases per 100,000 people [2]. Despite standard of care treatment of surgery followed by concomitant radiotherapy and chemotherapy [3], outcomes for GBM patients are extremely poor with a median survival of ~14 months [4,5]. Therefore, new treatment options are urgently required to improve patient outcomes. Agonists of the tumour necrosis factor (TNF)-related apoptosis-inducing ligand (TRAIL) receptors, TRAIL-R1 and TRAIL-R2, are one such potential treatment option. TRAIL receptor agonists, upon TRAILR1/R2 receptor aggregation, induce the extrinsic pathway of apoptosis exclusively in cancer cells whilst leaving normal cells intact [6,7,8]. Recently, second-generation TRAIL-receptor agonists were developed that show improved in vivo half times and cancer cell cytotoxicity [9,10,11]. Fc-scTRAIL is a second-generation TRAIL-receptor agonist, produced by fusion of a single-chain TRAIL (scTRAIL) trimer to the Fc region of an IgG, resulting in an overall hexavalent TRAIL-receptor agonist that potently engages TRAIL receptor-mediated apoptosis in a wide range of cancer cells [12,13,14,15,16]. The penetration of large biologics, such as TRAIL, into the CNS, is generally prevented by the presence of the blood–brain barrier (BBB), with approximately 0.1% of injected antibody doses reaching the brain parenchyma [17,18,19]. Whilst the BBB blocks almost all passive entry of biologics into the CNS, it actively transports select proteins and lipoproteins across the BBB by the process of receptor-mediated transcytosis (RMT) [20,21]. RMT is initiated by binding specific receptors on the BBB, invoking endocytosis of the cargo into a transport vehicle, whereupon the vesicle is trafficked to the opposite side of the BBB and exocytosed into the CNS parenchyma [17,18,22,23]. Many RMT targets at the BBB have been established, including the transferrin receptor (TfR) [21,24,25] low-density lipoprotein (LDL) receptor-related protein (LRP1) [26,27,28] and the insulin receptor [19,29]. Notably, despite the identification of RMT in the 1980s and the recent observation that entry of endogenous proteins into the CNS by RMT was much higher than originally thought [30], therapeutics based on the concept of RMT have yet to translate into clinically approved drugs.

Angiopep-2 (ANG2) is a small 19 amino acid peptide that was shown to induce LRP1-dependent RMT across the BBB [26,31] and it was used to increase CNS-penetrance of various cargo from small drugs to proteins to nanoparticle-based systems [32,33,34]. Recently, ANG2-based constructs have entered early phase I/II clinical trials, showing low toxicity (NCT01480583, NCT01967810, NCT02048059). In this work, we set out to explore whether ANG2-TRAIL fusion proteins can be generated without compromising the potency of the hexavalent TRAIL moiety to induce apoptotic cell death and if such a fusion protein can be transported across BBB endothelial cells through its ANG2 moiety.

## 2. Results

### 2.1. Design, Production and Purification of a CNS-Targeted Hexavalent TRAIL-Receptor Agonist

Second-generation hexavalent TRAIL fusion proteins have evolved as a potential treatment strategy against GBM [13,35] but the poor ability of biologics to cross the BBB is severely limiting to their clinical implementation. To address the complex problem of transporting high molecular weight TRAIL variants across the BBB, we fused TRAIL receptor agonists to ANG2, a synthetic peptide known to undergo transcytosis via the LRP1 receptor [26,31]. We included three functional units, the crystallisable fragment (Fc) domain of a human IgG including the hinge region, a single-chain format of trimeric TRAIL (scTRAIL) and ANG2. These units confer the final construct with dimerisation (Fc; Figure 1(Ai)), apoptosis induction (scTRAIL; Figure 1(Aii)) and CNS targeting (ANG2; Figure 1(Aiii)) properties. To induce effective apoptosis, we previously generated Fc-scTRAIL, consisting of two trivalent scTRAIL units dimerised via their fusion to the C-terminus of an Fc from human IgG1, creating the overall hexavalent TRAIL receptor agonists demonstrated to have improved cytotoxicity [12,13] (Figure 1(Bi,Ci)). We now also created a CNS-targeted hexavalent TRAIL-receptor agonist by fusing scTRAIL to the N-terminus of human IgG1 Fc and ANG2 to the C-terminus of Fc interspaced with a flexible glycine-serine linker (G_4_S)_2_ (Figure 1(Bii)). The resulting hexavalent TRAIL construct included two ANG2 moieties per molecule (Figure 1(Cii)). As an additional control, we generated scTRAIL-ANG2, a single ANG2 fused to the C-terminus of scTRAIL interspaced with a glycine-serine linker (G_2_SG_2_)_2_ (Figure 1(Biii,Ciii)). Notably, we created our TRAIL-ANG2 fusion proteins (Figure 1(Biii,Ciii)) with ANG2 on the C-terminus in accordance with previous studies [36,37]. To determine if C-terminal fusion was only a convention or serves functional purposes, we created two separate ANG2-positive control proteins by fusing ANG2 to the C- or N-terminal end of an Fcγ receptor binding (FcγR)-deficient mutant Fc (Fc^Δab^) [38], interspaced with a glycine-serine linker (G_4_S)_2_ (Figure 1(Biv,v,Civ,v)).

The proteins were produced in HEK293-6E cells, purified and analysed for purity, stability, aggregation and correct oligomerisation. In size-exclusion chromatography (SEC), we obtained a single peak of each protein with no unexpected aggregation, demonstrating a high purity and stability of the construct (Appendix A). In SDS-PAGE, we observed clear bands with no degradation and the proteins ran at their predicted molecular mass, in both reducing (monomeric) and non-reducing (dimeric) conditions (Appendix A). The presence of ANG2 in scTRAIL-Fc-ANG2 was also confirmed by MALDI-TOF mass spectrometry (Appendix A). Therefore, we designed and successfully purified fusion proteins and relevant control proteins to combine the potent and specific anti-cancer properties of hexavalent TRAIL-receptor agonists with enhanced CNS delivery properties of ANG2.

### 2.2. Fusion of ANG2 to Hexavalent TRAIL Maintains Its Potency in Inducing Apoptosis

First, we sought to confirm that the addition of ANG2 does not affect TRAIL-mediated activation of caspases. Thereby, we analysed caspase cascade signalling in the GBM cell line A172 and a well-characterised TRAIL-responsive human colon cancer cell line HCT116 in response to equimolar amounts of the TRAIL trimer, i.e., 1 nM hexavalent scTRAIL-Fc-ANG2 or 2 nM trivalent scTRAIL-ANG2. The hexavalent construct scTRAIL-Fc-ANG2 induced robust processing of the initiator procaspase-8, the downstream effector procaspase-3, and the caspase-3 substrate PARP. Processing of the caspases was inhibited by the pan-caspase inhibitor Q-VD-Oph in A172 and HCT116 cells (Figure 2A). The trivalent TRAIL construct, scTRAIL-ANG2, induced only minor processing of apoptosis signalling mediators in both cell lines. Having confirmed that ANG2-functionalised TRAIL-receptor agonists induce molecular hallmarks of apoptosis, we next determined cell death induction in A172 cells in response to Fc-scTRAIL, scTRAIL-Fc-ANG2 or scTRAIL-ANG2 by annexin V-PI staining and flow cytometry. We observed a strong response of cells to both hexavalent TRAIL constructs scTRAIL-Fc-ANG2 and Fc-scTRAIL in a dose-dependent manner (Figure 2B). Moreover, the EC_50_ values of Fc-scTRAIL (0.15 nM) and scTRAIL-Fc-ANG2 (0.11 nM) stayed in a similar range, demonstrating that the addition of ANG2 did not affect the TRAIL potency in killing cells. We also showed similar results in HCT116 cells using a crystal violet assay with EC_50_ values of 0.13 nM (Fc-scTRAIL) and 0.025 nM (scTRAIL-Fc-ANG2) (Figure 2C). In contrast, we observed poor responsiveness of A172 cells to the trivalent scTRAIL-ANG2 construct, with only a 30% loss of cell viability at the highest concentration of 4 nM (Figure 2B). This demonstrates that the enhanced clustering from hexavalent TRAIL-receptor agonists is essential for efficient apoptosis induction. Furthermore, hexavalent TRAIL constructs maintain their potency in inducing apoptosis after the addition of the ANG2 moiety.

### 2.3. Blood–Brain Barrier Cells Are Highly Resistant to TRAIL Treatment

TRAIL induces apoptosis by engaging with TRAIL receptors on the cell surface, therefore we quantified the number of surface TRAIL receptors on the human BBB endothelial cell line hCMEC/D3 and compared them to TRAIL-responsive HCT116 cells using flow cytometry. The hCMEC/D3 cells showed significantly lower expression of death receptors, TRAIL-R1 and TRAIL-R2, compared to HCT116 cells (Figure 3A). Importantly, these two receptors are the key mediators of downstream apoptosis signalling in response to TRAIL [39,40,41]. The decoy receptors, TRAIL-R3 and TRAIL-R4, were expressed at low levels in both cell lines with a marginal decrease in TRAIL-R4 in hCMEC/D3 cells compared to HCT116 cells (Figure 3A). This suggests that hCMEC/D3 cells and BBB endothelial cells may be more TRAIL-resistant than cancer cells. We therefore analysed apoptotic caspase signalling in hCMEC/D3 and the murine BBB endothelial cell line bEnd.3 for responses to hexavalent TRAIL by treatment with a low (0.1 nM) and a high (4 nM) concentration of Fc-scTRAIL for 6 h. As expected, even at the higher concentration of Fc-scTRAIL only residual accumulation of cleaved subunits was observed in hCMEC/D3 cells (Figure 3B), whereas no caspase processing was observed in bEnd.3 cells (Figure 3C). To confirm resistance of blood–brain barrier cells to TRAIL treatment, we determined the percentage of living hCMEC/D3 cells in response to Fc-scTRAIL, scTRAIL-Fc-ANG2 or scTRAIL-ANG2 by annexin V-PI staining and flow cytometry. We observed only residual loss in viability even at very high concentrations of 3 nM (Figure 3D). At the same concentration, nearly 100% cell death in A172 cells was achieved (Figure 2A). Furthermore, at the EC50 concentration for A172 cells (0.11 nM), hCMEC/D3 cells showed absolutely no discernible cell death (Figure 3D). The additional BBB endothelial cell line, bEnd.3, was also resistant to TRAIL-mediated viability loss (Figure 3E). Altogether, these data demonstrate that blood–brain barrier cells are resistant to TRAIL treatment, which corresponds to their low expression of the death receptors TRAIL-R1 and TRAIL-R2.

### 2.4. Binding of CNS-Targeted TRAIL Fusion Proteins to Blood–Brain Barrier Cells Is Predominantly TRAIL-Mediated

Having confirmed that BBB endothelial cells are resistant to TRAIL-mediated apoptosis, we next set out to characterise the modality of binding of TRAIL-ANG2 fusion proteins with blood–brain barrier cells. First, we sought to confirm the expression of the ANG2-target receptor LRP1 on human and mouse BBB endothelial cells. Surprisingly, western blot analysis and flow cytometry measurements demonstrated that hCMEC/D3 cells express very low levels of LRP1 compared to the known LRP1-expressing mouse embryonic fibroblasts (MEFs) [42] or bEnd.3 cells (Appendix A). Therefore, we conducted subsequent binding and transport studies in bEnd.3 cells. Given that BBB endothelial cells express TRAIL-receptors, albeit, at low levels, we initially set out to determine whether TRAIL-ANG2 fusion proteins preferentially bind to blood brain barrier cells via their TRAIL- or ANG2-targeting moieties. Hereby, we first incubated bEnd.3 cells with Fc-scTRAIL or scTRAIL-Fc-ANG2 for 2 h at 4 °C to prevent internalisation and then measured the binding using flow cytometry. To determine the nature of the binding, we also pre-incubated TRAIL constructs with a 100-fold molar excess of a soluble recombinant TRAIL receptor (TRAIL-R2-Fc), engineered by fusing the extracellular domain of TRAIL-R2 to an Fc, to block TRAIL-mediated binding to target cells. We observed dose-dependent binding of the fusion proteins to bEnd.3 cells, however, the binding was strongly inhibited when blocking TRAIL (Appendix A). This suggested that TRAIL-mediated binding dominated under these assay conditions. Given the reported low affinity (313 nM) of ANG2 for LRP1 [43], we reasoned that ANG2-binding to the cells at 4 °C may be too low for specific robust detection of surface binding. Indeed, as expected, the binding of various ANG2-positive control proteins, FLAG-ANG2, FITC-ANG2 or FITC-scrambled ANG2 (FITC-scrANG2) to bEnd.3 cells at 4 °C was not detectable (Appendix A). Moreover, the binding of FITC-ANG2 was not increased compared to scrambled control, suggesting the signal was predominantly due to non-specific interaction with the FITC-label. Therefore, we switched to an immunostaining-based analysis where we simultaneously observed both cell surface binding and uptake of constructs at 37 °C. Hereby, we incubated the cells with 50 nM of Fc^Δab^-ANG2, ANG2-Fc^Δab^, Fc-scTRAIL or scTRAIL-Fc-ANG2 for 15 min or 30 min and imaged cells using confocal microscopy. We observed specific signals from positive control proteins Fc^Δab^-ANG2 and ANG2-Fc^Δab^ to bEnd.3 cells after 15 min or 30 min incubation, as quantified by either counting of puncta (Figure 4A,B) or total cell fluorescence intensity (Figure 4C,D). Notably, binding was independent of whether ANG2 was fused to the N- or C-terminus of the protein. Strikingly, we observed a marked fold increase of between 2.9 and 4.5 in the intensity of binding of TRAIL-based constructs, Fc-scTRAIL and scTRAIL-Fc-ANG2, as compared to the ANG2-only positive controls (Figure 4C,D), in line with the flow cytometry experiments (Appendix A). Moreover, signals from the TRAIL-based constructs were highly diffuse across the entire cell, whilst ANG2-only constructs were predominantly localised to the perinuclear region (Figure 4A,D), in line with the spatial expression pattern of its target receptor, LRP1 [44] (Appendix A). TRAIL-R2-mFc^LALA^, a fusion of the extracellular domain of the human TRAIL-R2 and the Fc-receptor binding mutant murine Fc (mFc^LALA^), was used to prevent the non-specific signal from anti-human Fc-based detection and engagement of Fc-receptors. Blocking of TRAIL with a 100-fold molar excess of TRAIL-R2-mFc^LALA^, brought the binding back to the level of ANG2-only control proteins and returned the spatial distribution to the perinuclear region (Figure 4C,D). Taken together, these data demonstrate that ANG2 is functional within fusion proteins, but that despite low expression of TRAIL receptors on blood–brain barrier cell lines, masking or absence of TRAIL or TRAIL receptors is required for effective detection of ANG2-mediated binding.

### 2.5. CNS-Transport of TRAIL-Fusion Constructs

Having established that scTRAIL-Fc-ANG2 binds both via ANG2 and TRAIL moieties to BBB endothelial cells, we next set out to characterise transcytosis of this fusion protein across BBB endothelial cells. We established an in vitro blood–brain barrier model by growing bEnd.3 cells to a confluent layer on a transwell (Figure 5A) and obtained a steady barrier with a transendothelial electrical resistance (TEER) of 17 Ω·cm^2^ (Appendix A). We then placed 20 nM of Fc^Δab^-ANG2, ANG2-Fc^Δab^, Fc-scTRAIL and scTRAIL-Fc-ANG2 in the apical compartment with or without 30 min pre-incubation with a 100-fold molar excess of soluble TRAIL-R2-mFc^LALA^. After 1 h incubation at 37 °C, we determined the concentration of fusion proteins in the apical and basolateral compartments using quantitative sandwich ELISA. ANG2-only positive control proteins, ANG2-Fc^Δab^ and Fc^Δab^-ANG2, were both detected after 1 h in the basolateral compartment (Figure 5B). In line with our finding that ANG2 binding was independent of whether ANG2 was fused to the N- or C-terminus of the protein, successful transport was also observed for both conformations of the ANG2-controls. On the contrary, scTRAIL-Fc-ANG2 was only detectable in the basolateral compartment after blocking of TRAIL through TRAIL-R2-mFc^LALA^. As expected, Fc-scTRAIL was not transported to the basolateral compartment, confirming that the ability to be transported is strictly ANG2-dependent. Overall, this indicates that binding to TRAIL receptors interferes with ANG2-mediated transport of scTRAIL-Fc-ANG2 across BBB endothelial cells, but that TRAIL-based biologics can be transported when interfering with TRAIL/TRAIL-R interactions on the apical side. This provides a rationale for TRAIL/TRAIL-R masking strategies outside of the CNS to allow for the transport of this potent apoptosis inducer into the brain.

## 3. Discussion

TRAIL-based therapeutics have shown great potential in pre-clinical studies as a novel approach for the treatment of GBM [45,46,47] but the BBB prevents the entry of TRAIL-based biologics from effectively reaching GBM tumours. In this paper, we demonstrate that ANG2-functionalised hexavalent TRAIL receptor agonists retain their potency in killing GBM cells, that BBB endothelial cells are resistant to TRAIL and able to transport large ANG2-based fusion proteins across the BBB. However, transport of ANG2-TRAIL fusion proteins is only possible when the TRAIL binding to BBB endothelial cells is blocked.

Previously, it was demonstrated that TRAIL receptor agonists have broad efficacy against GBM cells alone or in combination with sensitisers [35,45,46,47,48,49]. Beyond GBM, the CNS is also a frequent secondary site for many cancer metastases, including lung cancer, breast cancer and melanoma, consequently leading to lower treatment responses and poor patient outcomes [50,51]. Therefore, a CNS-targeted therapeutic variant of TRAIL, which shows broad anti-cancer efficacy in various cancer types [15,16,52,53] would be of considerable clinical interest as an anti-cancer agent. Furthermore, recent studies have suggested that endogenous TRAIL plays an important role in immune modulation in multiple sclerosis [54], suggesting a CNS-targeted TRAIL variant could also be used in the treatment of multiple sclerosis and other inflammatory CNS disorders [55,56]. Taken together, this demonstrates that a CNS-targeted therapeutic variant of TRAIL would have broad therapeutic potential and is of considerable clinical interest.

The clinical deployment of TRAIL variants for CNS disease is greatly impeded by the poor penetration of large molecular drugs into the CNS [17]. We took advantage of a widely exploited mechanism for the delivery of therapeutics across the blood–brain barrier by targeting the transport pathway of receptor-mediated transcytosis [21,57,58]. Whilst a number of RMT-inducing candidates have been described, we deployed the small peptide ANG2 to increase CNS-penetration of TRAIL variants due to its established efficacy [26,31,43], safety in patients [59], broad species specificity and ease of inclusion of a 19 amino acid peptide within fusion proteins, which makes it an optimal choice for such an exploratory study. In initial apoptosis studies, we demonstrated that the addition of ANG2 maintained the potency of hexavalent TRAIL in inducing apoptosis within the GBM cells, as determined by Annexin V-PI staining and efficient cleavage of the key apoptotic regulators, pro-caspase 8, pro-caspase 3 and PARP. Similar to previous studies, the higher clustering potential of hexavalent scTRAIL was strictly necessary for efficient apoptosis induction in GBM cells as this is required for efficient engagement of TRAILR2 [10,12,13,14]. Importantly, hexavalent TRAIL receptor agonists based on a similar format, ABBV-621, are currently undergoing clinical trials (NCT03082209). Thus, hexavalent TRAIL could serve as a promising therapeutic agent in the treatment of GBM patients. In contrast to cancer cells, BBB endothelial cells were largely resistant to hexavalent TRAIL receptor agonists even at high concentrations, in accordance with the concept that TRAIL-mediated apoptosis occurs almost exclusively in cancer cells [6,60]. We demonstrated that levels of TRAIL receptors in the BBB endothelial cell line hCMEC/D3 were considerably reduced compared to cancer cells. The receptor numbers in these cells were also generally lower than published levels in a wide range of cancer cell lines [15,16]. This is in line with previous work demonstrating that endothelial cells do express TRAIL receptors but are resistant to TRAIL-mediated apoptosis [61,62,63]. Previous studies have also shown that endothelial cells of the BBB are highly resistant to extrinsic apoptosis, due to the activity of the pro-survival factors TAK1 and NEMO [64]. This, together with our findings of low TRAIL-receptor expression, demonstrates that there are multiple factors contributing to high TRAIL resistance in BBB endothelial cells.

Our combined binding data from flow cytometry and immunostaining demonstrated that ANG2-based constructs bind to blood–brain barrier cells. In the transport assay, this was sufficient for the transport of ANG2-only constructs across BBB endothelial cells. We also demonstrated that ANG2 was active within fusion proteins regardless of whether it was fused to the N- or C-terminus of the protein. Whilst other studies of ANG2 use C-terminal labelling, our data suggest this is not an absolute requirement for ANG2 function [36,37]. Our transport assays were performed using b.End3 cells grown on a transwell insert. We reported a TEER value for this setup of 17 Ω·cm^2^. Previous TEER values reported for b.End3 cells vary widely from 15–140 Ω·cm^2^ [65,66,67]. Whilst our reported TEER values are on the lower side of this range, they are in line with what was previously reported and we observed no apparent effect from passive diffusion at the time step chosen. Applying this in vitro BBB model, we achieved transport rates of (0.1–0.3 pmoles/cm^2^/h). Whilst these are lower than the reported rates for ANG2 crossing bovine brain endothelial cells (~6 pmoles/cm^2^/h) [31], this is likely due to the significantly lower concentration of protein added to the apical side in our transwell experiments. Due to the lack of LRP1 expression on hCMEC/D3 cells, we were unable to extend our transport findings to human BBB endothelial cells. Whilst angiopep-2 was shown to be active in different species, including human clinical trials (NCT03613181), further investigation is required to ascertain how these constructs behave in human models [26,31,36].

Our key finding was that the presence of TRAIL interfered with ANG2-mediated transport, despite the expression of TRAIL receptors being low. As binding rates are dictated by individual affinities, receptor availability and avidity effects, this suggests that either the receptor levels of LRP1 are lower than for TRAIL receptors or that TRAIL has a significantly higher binding rate for its receptors than ANG2, potentially due to higher individual affinity or overall avidity. Importantly, recent studies have shown that reduced overall affinity binding is beneficial for efficient RMT, whether by lowering the affinity itself or by reducing the avidity of binding [68,69,70,71,72,73]. This is apparently due to the redirection of cargo to lysosomal compartments instead of being trafficked across the cell [69,70,74,75]. The reported low affinity of ANG2 to its target receptor LRP1, 330 nM [43], suggests that although it was not deliberately engineered as a reduced affinity binder, it operates in such a manner. Indeed, the precursor protein, angiopep-1, shows greater total brain distribution but has greater accumulation within capillary fractions suggesting similar lysosomal sorting occurs for LRP1-mediated transcytosis [31]. Hexavalent TRAIL receptor agonists, on the other hand, bind to cells at sub-nanomolar concentrations, due to the combined high affinity and avidity of the hexavalent agonist [9,12,76,77]. This several orders of magnitude higher affinity of TRAIL, together with the expression of TRAIL receptors at BBB endothelial cells, would potentially explain why we found that TRAIL-ANG2 fusion proteins bound BBB endothelial cells in a predominantly TRAIL-mediated manner, consequently resulting in poor transport across the BBB. Whilst such an effect has not been shown in BBB delivery before, similar effects were demonstrated in bispecific anti-tumour therapeutics, where reducing the affinities of one arm alters drug disposition and can improve therapeutic efficiency. This is a particularly important consideration in the development of future therapeutics based on the concept of reduced affinity brain shuttles, the selection of therapeutic targets that do not interfere with BBB endothelial cell binding and transport. Importantly, quantitative proteomics data in isolated brain microvessels have demonstrated that LRP1 is expressed at lower levels when compared to other RMT targets such as the transferrin receptor in both mice and humans [78,79,80]. The greater levels of receptors for other targets may help to offset the reduced affinity of binding. Alternatively, in the future, an adaptation of the high-affinity TRAIL moiety could be utilised to facilitate better transcytosis. This can be achieved by mutations reducing affinity to the TRAIL receptors or by lowering the avidity with trimeric TRAIL receptor agonists that are selectively clustered in situ at cancer cells. Importantly, various mutations that reduce TRAIL affinity to its receptors were previously reported [81] and enhanced clustering of TRAIL receptors at target cancer cells by dual targeting against epidermal growth factor receptor (EGFR) was also established [12,82] suggesting these could be viable future strategies to reduce TRAIL affinity and therefore increase its CNS-delivery. Alternatively, a TRAIL- or TRAIL-R blocking strategy could be employed either using systemic blocking peptides, as demonstrated in this paper, or by including interfering moieties within the fusion-protein itself that are selectively cleaved off within the CNS, which will then allow the efficient ANG2-mediated transport of TRAIL-fusion proteins into the CNS. Overall, our data demonstrate that the high-affinity binding of TRAIL therapeutics interferes with Angiopep-2-mediated across BBB endothelial cells. Our findings emphasise that BBB expression of high-affinity therapeutic targets can interfere with transport processes and is an essential consideration when designing CNS-targeted therapeutics based on reduced affinity RMT.

## 4. Materials and Methods

### 4.1. Reagents and Antibodies

The peptides FITC-ANG2 (FITC-Ahx-TFFYGGSRGKRNNFKTTEEY), FITC-scrANG2 (FITC-Ahx-NSFEGTGGEYFTYRKRNFK) and FLAG-ANG2 (FLAG- TFFYGGSRGKRNNFKTTEEY) were purchased from Peptides and Elephants (Brandenburg, Germany). Hoechst 33342 was obtained from Thermo Fisher Scientific (Waltham, MA, USA). QIFIKIT was obtained from Dako (K0078, Glostrup, Denmark). Annexin V-EGFP was produced in-house. PI was obtained from Sigma Aldrich (Munich, Germany). For flow cytometry, the following antibodies were used: primary antibodies: mouse anti-TRAILR1/TNFRSF10A (MAB347, 1:100), mouse anti-TRAILR2/TNFRSF10B (MAB6311, 1:100), mouse anti-TRAILR3/TNFRSF10C (MAB6302, 1:100), mouse anti-TRAILR4/TNFRSF10D (MAB633, 1:100), purified mouse IgG1 (1:100) and purified mouse IgG2b (1:100), purchased from R&D Systems (Wiesbaden-Nordenstadt, Germany). Secondary antibodies: anti-FLAG-PE (130-101-577, 1:200, Miltenyi Biotec, Bergisch Gladbach, Germany), goat anti-mouse FITC (F0479, 1:50, Dako, Glostrup, Denmark). Anti-FLAG M2 affinity gel (A2220, Sigma-Aldrich, Munich, Germany), FLAG peptide (EPO174, Peptides and Elephants, Brandenburg, Germany) were used for protein purification. The antibodies used for immunocytochemistry were, primary antibodies: anti-FLAG rabbit (F7425, 1:100, Merck KGaA (Darmstadt, Germany), anti-FC-PE (109-115-098, 1:50, Jackson ImmunoResearch, Cambridgeshire, UK). Secondary antibodies: goat anti-rabbit 131 Alexa Flour 647 IgG (H + L) (A-21245), goat anti-mouse Alexa 132 488 IgG (H + L) (A-11029), goat anti-rabbit Alexa 488 IgG 133 (H + L) (A-11008), all purchased from Thermo Fisher Scientific (Waltham, MA, USA). The following antibodies were used for western blotting: mouse caspase 8 (IC12), rabbit caspase 8 (D35G2), rabbit procaspase 3 (9662), rabbit PARP (9542) mouse α-tubulin (DM1A), rabbit β-actin (4967), all purchased from Cell Signaling Technology (Danvers, MA, USA) and mouse PARP (4C10-5) from BioLegend (San Diego, CA, USA). For ELISA, the following antibodies were used, goat human IgG (Fc) specific-alkaline phosphatase from Sigma-Aldrich (Munich, Germany), mouse IgG (Fc) CH2 Domain- HRP from Bio-Rad Antibodies (Hercules, CA, USA).

### 4.2. Generation of Constructs

The plasmid pSecTagA-scTRAIL(281-G118)-(GGSGG)4-Fc was used for the generation of scTRAIL-Fc-ANG2 and sc-TRAIL-ANG2, which were obtained by PCR with respective primers and then digested with restriction enzymes Age1, EcoR1 and cloned into pSecTag-AL1 vector to obtain pSecTagAL1-Flag-scTRAIL-Fc-ANG2 and pSecTagAL1-FLAG-scTRAIL-ANG2. For the generation of mutant Fc^Δab^-ANG2 plasmids, FcΔab was cut from pAB1-L1-Dbhu225x3-43-CH1-FcΔab and then cloned into pSecTag-AL1 with respective primers to obtain pSecTagAL1-FcΔab-Angiopep2 and pSecTagAL1-Angiopep2-FcΔab. The TRAIL-R2-Fc was cloned using the extracellular domain of TRAIL-R2 and produced as previously described [83]. The TRAIL-R2-mFc^LALA^ was cloned using the extracellular domain of TRAIL-R2 and fused with the primers to murine Fc part containing the PGLALA-mutation vector.

The following primers were used for the generation of constructs: scTRAIL-Fc-ANG2: 5′-

TTTGAATTCTCAGTATTCTTCCGTCTTAAAATTGTTCCTCTTGCCCCTAGACCCACCGTAGAAAAAAGTAGAACCGCCTCCTCCACCTGATCCGCCACCTCCTTTACCCGGAGACAGGG-3′ (forward), 5′-AAAACCGGTGACTACAAAGACG-3′ (backward); scTRAIL-ANG2: 5′-

TTTGAATTCTCAGTATTCTTCCGTCTTAAAATTGTTCCTCTTGCCCCTAGACCCACCGTAGAAAAAAGTGCCGCCAGATCCGCC-3′ (forward), 5′-

AAAACCGGTGACTACAAAGACG-3′ (backward); Fc^Δab^-ANG2: 5′-

TTTTTGAATTCTCAGTATTCTTCCGTCTTAAAATTGTTCCTCTTGCCCCTAGACCCACCGTAGAAAAAAGTAGATCCACCGCCACCTGATCCGCCACCTCCTTTGCCGGGGGACAGGC-3′ (forward), 5′-AAAAAACCGGTGACAAGACCCACACCTGTC-3′ (backward), ANG2-Fc^Δab^: 5′-TTTTTGAATTCTTATTTGCCGGGGG-3′ (forward), 5′-

AAAAAACCGGTACTTTTTTCTACGGTGGGTCTAGGGGCAAGAGGAACAATTTTAAGACGGAAGAATACGGAGGTGGCGGATCAGGTGGCGGTGGATCTGACAAGACCCACACCTGTC-3′ (backward); TRAIL-R2-mFc^LALA^: 5′-

TTTTGAAGACCCCTCCTGAGAGAGAACAGGGAGAG-3′ (forward), 5′-

AAAAGAAGACGGACCGGTGAGTCTGCTCTGATCACCC-3′ (backward).

### 4.3. Production and Purification of the Recombinant Proteins

TRAIL constructs were produced in HEK293-6E cells with TRAIL-R2 knockout, and transfected cells were cultured in F17 medium (A1383503, Thermo Fischer Scientific, Waltham, MA, USA) supplemented with 50 μM ZnCl_2_ and tryptophan (TN1, Organotechnie S.A.S., La Courneuve, France). Later, the supernatants were collected and incubated with anti-FLAG M2 affinity gel overnight on a roller mixer at 4 °C. After washing the column with 1x TBS, the beads were added and eluted with 100 μg/mL FLAG peptide. For Fc fusion ANG2, TRAIL-R2-Fc, TRAIL-R2-mFc^LALA^ constructs, the transfection was carried out in HEK293 cells cultured in F17 medium. The supernatants were incubated with protein A Sepharose beads overnight on a roller mixer at 4 °C and the elution was carried out with protein A elution buffer (Glycine-HCl pH 3.5). For all proteins produced, dialysis was carried out overnight in 1x PBS and proteins were concentrated with Vivaspin 20 centrifugal concentrators (VS2001/VS2031, Sartorius, Göttingen, Germany). Protein concentrations were measured using a Nanodrop instrument (Thermo Fischer Scientific, Waltham, MA, USA), based on calculated molecular weights and extinction coefficients. As a further purification step, size exclusion fast protein liquid chromatography was performed using Superdex 200 10/300 GL column (PBS as mobile phase, flow rate of 0.5 mL/min). Subsequently, produced proteins were analysed by SDS page and HPLC size exclusion chromatography using Phenomenex Yarra 3 μm SEC-2000 or -3000 column (00H-4512-K0 or 00H-4513-K0, Phenomenex, CA, USA), a Waters 2695 HPLC, and a mobile phase consisting of 0.1 mol/L Na_2_HPO_4_/NaH_2_PO_4_, 0.1 mol/L Na_2_SO_4_, pH 6.7 at a flow rate of 0.5 mL/min.

### 4.4. Cell Culture

The cell lines A172, bEnd.3 and HCT116 were obtained from ATCC (Manassas, WV, USA), HEK293-6E from National Research Council of Canada (Ottawa, ON, Canada), hCMEC/D3 from Merck Millipore (Darmstadt, Germany), MEF WT from Institute of Cell Biology and Immunology (kindly provided by Dr. rer. nat. Kornelia Ellwanger). The cells were grown as per supplier’s instructions. The hCMEC/D3 cells were grown in Endothelial Cell Growth medium MV with supplements including 0.05 mL/mL of Fetal Calf Serum, 0.004 mL/mL of Endothelial Cell Growth Supplement, 10 ng/mL of Epidermal Growth Factor, 90 μg/mL of Heparin, 1 μg/mL of Hydrocortisone and 1 ng/mL of basic Fibroblast Growth Factor (C-39221, Promo cell, Heidelberg, Germany). The flasks were coated with 10 μg cm^−2^ Rat Tail Collagen I (A10483-01, Thermo Fischer Scientific, Waltham, MA, USA) in PBS for 1 h at 37 °C. The bEnd.3, A172 cells were cultured in DMEM (41965-039, Thermo Fischer Scientific, Waltham, MA, USA) and HCT116 were cultured in RPMI (21875–034, Thermo Fischer Scientific, Waltham, MA, USA) with 10% FCS (P30-3302, PAN Biotech, Aidenbach, Germany). All cells were maintained in a humidified incubator at 37 °C with 5% CO_2_. All cell lines were tested for absence of mycoplasma infection.

### 4.5. Receptor Quantification

For quantifying cell surface receptors, cells were harvested (100,000 cells/well) and resuspended in cold PBA [1x PBS, 0.05% BSA (Bio&Sell, Feucht, Bavaria, Germany) and 0.02% Sodium Azide (422.1, Carl Roth, Karlsruhe, Germany)]. Then the cells were incubated in primary antibody for 1 h on ice, followed by washing with PBA and incubation with secondary antibody for 1 h on ice. Subsequently, the cells were washed with PBA and measured using a MACSQuant analyser 10 flow cytometer against DAKO QIFIKIT beads as per manufacturer’s instructions. IgG1 (anti-TRAILR1, anti-TRAILR3, anti-TRAILR4) and IgG2b (anti-TRAILR2) were used as isotype controls.

### 4.6. Cell Binding Assays

Cells (50,000 cells/well) were harvested and resuspended in cold PBA (1xPBS, 0.05% BSA and 0.02% Sodium Azide). The cells were incubated with different constructs in serial dilution for 2 h on ice. For blocking experiments, cells were pre-incubated for 30 min with constructs in which the TRAIL moieties were blocked with TRAIL-R2-Fc or TRAIL-R2-mFc^LALA^. After incubation, cells were washed with PBA and incubated for 2 h with the secondary antibody depending on the detection moiety of the constructs. The binding was measured by flow cytometry and the median fluorescence intensities were obtained for each construct. For display purposes, we normalised FITC-ANG2 and FITC-scrANG2 binding intensity to the lowest concentration of scrANG2.

### 4.7. Cell Death Assays

Cells were grown in 96-well flat-bottom plates for 24 h and incubated with the constructs in serial dilution. After 24 h, floating cells were collected while adherent cells were collected by trypsinisation with trypsin/EDTA (Thermo Fischer Scientific, Waltham, MA, USA) and stained with Annexin V/PI for 10 min at room temperature. For hCMEC/D3 and A172 cells, cell death was measured by flow cytometry. The percentage of living cells was defined as the percentage of AnV^−^ PI^−^ cells normalised against untreated control cells.

### 4.8. Crystal Violet Assay

10,000 cells/well were seeded in 100 μL medium in 96-well plate (F bottom) and grown for 24 h at 37 °C, 5 % CO_2_. A total of 200 μL of serial diluted proteins were added to the cells and were incubated for 16 h at 37 °C, 5% CO_2_. After incubation for 16 h, the cells were washed with PBS and 50 μL/well of crystal violet staining solution was added and incubated for 15 min at RT. Staining was removed and the cells were left to dry overnight at RT. A total of 100 μL of methanol was added per well and the absorption was measured at 570 nm on a microplate reader (SPARK, Tecan, Männedorf, Switzerland).

### 4.9. Mass Spectrometry

scTRAIL-Fc-ANG2 was loaded on Bolt 4–12% Bis-Tris Plus pre-cast gels and was run in a Bolt Mini tank with Bolt MES SDS running buffer purchased from Thermo Fischer Scientific (Waltham, MA, USA). The gel was washed with Milli-Q water to remove the buffer and then stained with instant Coomassie blue (11022018, Expedeon, Heidelberg, Germany). After staining, the gel was again washed with Milli-Q water. The stained protein bands were cut, washed with Milli-Q water and then centrifuged for 1 min at 11,000 rpm. To the gel pieces, 200 μL of 50 mM NH_4_HCO_3_ (09830-500G, Sigma Aldrich, Munich, Germany) and 100% acetonitrile (4722.1, Carl Roth, Karlsruhe, Germany) 1 + 1 (*v*/*v*) was added and incubated in the shaker for 15 min. The solution was discarded and the gel was incubated with 50 μL of 100% acetonitrile for 5 min for dehydration. Once the gel was shrunk, acetonitrile was removed and 50 mM of NH_4_HCO_3_ was added to the gel and incubated for 5 min for rehydration. Then an equal volume of acetonitrile was added and incubated for 15 min while shaking. The remaining solution was removed and the gel was air-dried in a vacuum centrifuge. For the in-gel digestion, the air-dried gel particle was digested with a solution of 5 ng/mL Trypsin (sequencing grade modified Trypsin, Promega, Walldorf, Germany), 50 mM NH_4_HCO_3_ and 100% acetonitrile and incubated overnight at 37 °C. The supernatant was removed and 25 μL of TA20 (20% acetonitrile and 0.1% of Trifluoroacetic acid, TFA (P088.2, Carl Roth, Karlsruhe, Germany) was added. The gel particles were ultrasonicated (USR32H, Merck eurolab, Lutterworth, UK) for 5 min and incubated for 30 min. The supernatant was collected and the steps were repeated with TA50 (50% acetonitrile and 0.1% of TFA). The supernatants were combined and dried for 1 h in a vacuum centrifuge. The dried pellet was dissolved in 10 μL of 0.1% TFA. A 1 µL of the sample was added to the MALDI target plate (MTP Anchor Chip 384TF, Bruker Daltronics, Billerica, MA, USA). After drying 1µL of Matrix solution (0.7 mg/mL a-Cyano-4-hydroxycinnamic acid (39468-10x10MG, Sigma Aldrich, Munich, Germany) was dissolved in a solvent mixture containing 85% acetonitrile, 15% H_2_O, 0,1% TFA and 1 mM NH_4_H_2_PO_4_ was spotted above the sample. For peptide calibration 0.5 µL Peptide Calibration Standard II (222570, Bruker Daltronics, Billerica, MA, USA) was spotted on the same target plate. The plate was inserted in the MALDI-TOF machine (Bruker^TM^ Autoflex Speed MALDI TOF, Billerica, MA, USA) and the samples were analysed using (Flex Control and Flex Analysis) software.

### 4.10. Immunocytochemistry

Cells were grown on coverslips coated with 2.5 μg/mL Collagen R solution (08-115, Sigma-Aldrich, Munich, Germany). Once the cells were confluent, they were incubated with the constructs for different times at 37 °C. After the incubation, cells were washed with PBS and fixed with 4% paraformaldehyde (sc-281692, Santa Cruz Biotechnology Inc., Santa Cruz, CA, USA) for 30 min. Thereafter, cells were permeabilised with Triton X-100 in PBS for 10 min and blocked with 4% BSA. The cells were incubated with primary antibodies for 1 h at RT and diluted in PBS followed by washing and incubation with 4 μg/mL secondary antibodies for 45 min at RT. After washing, cells were mounted with Fluoromount-G. Images were acquired on a confocal laser scanning microscope (LSM 710, Carl Zeiss, Oberkochen, Germany) equipped with a Plan-Apochromat 63x/1.4 Oil objective. DAPI was excited with a 405 nm diode laser, its emission was detected from 410–490 nm. PE was excited with a 561 nm DPSS laser, its emission was detected from 553–660 nm. All images were subjected to identical post-imaging processing for comparability, including linear adjustments to brightness/contrast and maximum intensity projections using the ZEN black software version 2.1 (Carl Zeiss). Images were quantified using CellProfiler version 3.1.8 [84]. In brief, cells were segmented and speckles in the PE channel under the cell mask were counted. Additionally, the mean intensity in the red channel was measured for each cell.

### 4.11. Western Blotting

For protein extraction, the cell pellets were resuspended in lysis buffer and incubated for 15 min on ice, then the samples were centrifuged for 15 min at 16,000 g and the supernatant was transferred to new reaction tubes. Protein concentrations were quantified by Bradford assay. The protein concentration was calculated and then prepared with ddH2O and 5x loading buffer with a final concentration of 1–2 μg/μL protein. The samples were incubated for 5 min at 95 °C on a heat block (HBT-1-131, Haep Labor Consult, Bovenden, Germany) and frozen at −20 °C. For the SDS PAGE, the samples and the molecular weight marker were loaded into Bolt 4–12% Bis-Tris Plus Gel 1.0 mm × 15 well or Bolt 4–12% Bis-Tris Plus Gel 1.0 mm × 17 well with Bolt MES SDS Running Buffer (20X) placed on Bolt Mini Gel Tank chamber (Thermo Fisher Scientific). The proteins were separated in the gel at 150 V for about 40 min. The separated proteins were transferred to a nitrocellulose membrane using the iBlot 2 Dry Blotting System (transfer settings: 20 V; 7 min, Thermo Fisher Scientific Inc.). The membranes were washed and incubated for at least 1 h with blocking reagent (diluted 1:10 in TBST, Roche Diagnostics, Basel, Elveția) washed three times with TBST for 10 min and incubated with the primary antibody (diluted in TBST with blocking reagent (1:20) and 0.02% (*w*/*v*) NaN3)) overnight at 4 °C or for 1 h at RT. After three more washing steps with TBST, membranes were incubated with a horseradish peroxidase (HRP) coupled secondary antibody (diluted in TBST with blocking reagent (1:20) 1 h at RT. The membranes were washed and prepared for detection. For the detection process, the membranes were incubated with an HRP substrate (SuperSignal West Pico ECL Substrate/SuperSignal West Dura Extended, Pierce Protein Research Products; Luminata Forte Western HRP Substrate from Merck Millipore (Darmstadt, Germany) and detected with Amersham™ Imager 600 (GE Healthcare Bio-Sciences, Pittsburgh, PA, USA).

### 4.12. MTT Assay

bEnd.3 cells seeded in a 96-well plate (F bottom) and let to attach and grow for 24 h. Then, the cells were treated with scTRAIL-Fc-ANG2, scTRAIL-ANG2 and IZI1551 in serial dilution for 24 h. After the treatment, 20 μL of MTT solution (5 mg/mL) were added to each well including the wells without cells. The plate was incubated for 3–4 h in the incubator at 37 °C. Later, the medium was aspirated and 200 μL of methanol was added to each well and incubated for 10 min in the incubator. The absorbance was measured at a wavelength of 570 nm with a Tecan plate reader (Infinite M200 or SPARK, Tecan, Meannedorf, Switzerland).

### 4.13. Blood–Brain Barrier Transwell Setup and Transport Assay

The bEnd.3 cells (30,000 cells/well), were seeded in 24-well transwell plate (0.33 cm^2^ surface area, 0.4 μm pore size, Sigma-Aldrich, St. Luis, MO, USA) with their respective medium. Transendothelial electrical resistance (TEER) was measured every day for 6–7 days using EVOM 2/STX2 electrode (World Precision Instruments, Sarasota, FL, USA) to check the integrity of barrier cells. The medium was changed once in two days. Two negative controls without the cells were maintained to subtract from the resistance of the samples. Once a stable TEER value was reached (plateaued for 3 consecutive days), the cells were prepared for the transport experiment. The cells were preincubated for 1 h with 50 μM QVD diluted in medium. After that, the cells were washed with sterile PBS. A total of 20 nM of Fc^Δab^-ANG2, ANG2-Fc^Δab^, scTRAIL-Fc-ANG2 and Fc-scTRAIL diluted in BSA (2%) were added to the wells with cells and incubated at 37 °C. After 1 h, the samples from the top and bottom were taken and were measured using ELISA. The time point of 1 h was used to allow efficient detection of basolateral protein as determined in a pre-experiment. The top well samples were diluted 1:20 for the measurement.

### 4.14. ELISA for the Transport Measurement

The ELISA plates from Greiner Bio-One (Frickenhausen, Germany) were coated with anti-human IgG (Fc) diluted (1:1000) in ELISA coating buffer and left overnight at 4 °C. The next day, ELISA plates were washed with ELISA washing buffer three times and blocked with 2% BSA in PBS for 1 h. Plates were washed with washing buffer five times. A total of 100 μL of the samples from the transwell and also the samples for the standard were added to the ELISA plate and incubated for 2 h at RT. The plates were again washed with washing buffer five times. The plates were incubated with the detection antibody anti-human IgG (Fc) CH2 Domain: HRP diluted (1:500) in 2% BSA for 2 h at RT. The plates were washed seven times with washing buffer and then incubated with TMB substrate for 5 to 10 min. After that, to stop the reaction, the ELISA stop solution was added. A standard curve was established for each construct and the transported proteins in the transwell were determined by interpolation from the standard curve and under consideration of the dilution factor. The absorbance was measured at a wavelength of 450 nm with a Tecan plate reader (Infinite M200 or SPARK, Tecan, Meannedorf, Switzerland).

### 4.15. Statistics

GraphPad Prism 7 (GraphPad Software, San Diego, CA, USA) was used to analyse all data. Statistical significance of difference between groups was performed by indicated significance test. Parametric tests were performed only after confirming normal distribution of the data using the D’Agostino and Pearson omnibus normality test. For receptor quantification, statistical analysis was performed on log-transformed values, after verification of log-normal distribution by D’Agostino and Pearson omnibus test. Significance levels were denoted with asterisks: * = *p* ≤ 0.05; ** = *p* ≤ 0.01; *** = *p* ≤ 0.001 **** = *p* ≤ 0.0001.

## Figures and Tables

**Figure 1 molecules-26-07582-f001:**
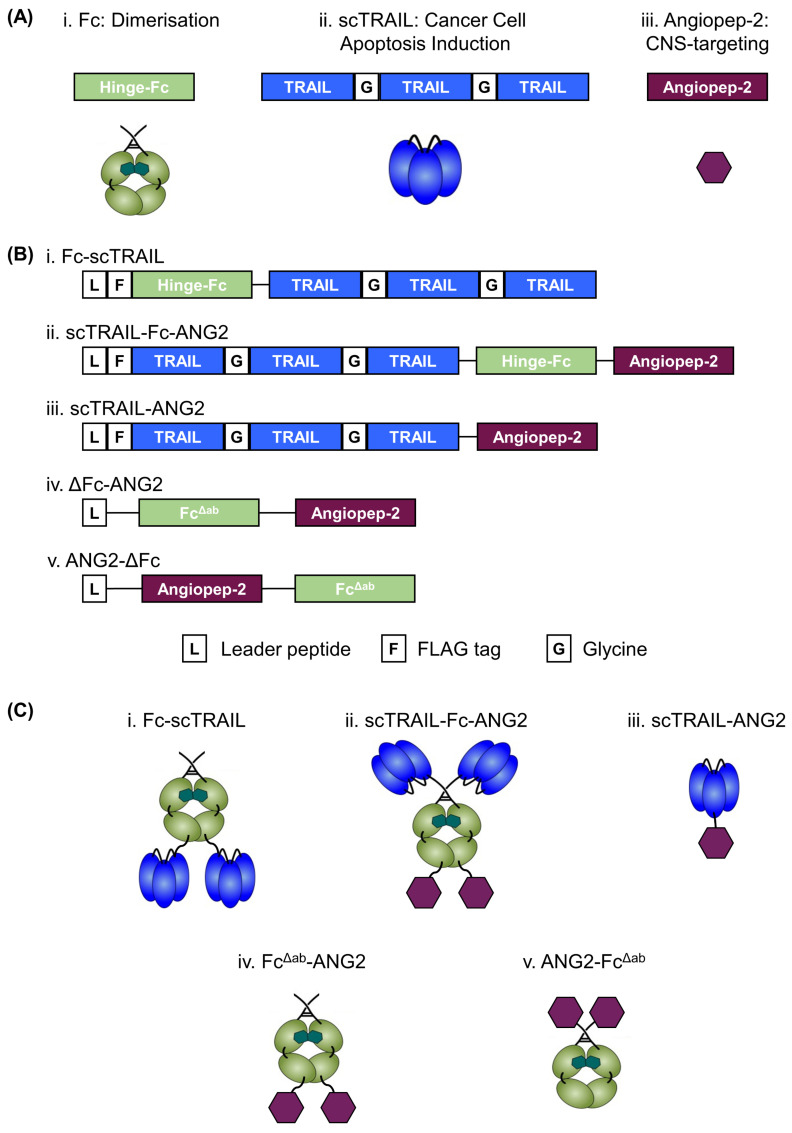
Designing a CNS-targeted TRAIL-receptor agonist. (**A**) Functional units, (**B**) composition and (**C**) schematic assembly of CNS-targeted scTRAIL variants and relevant control proteins.

**Figure 2 molecules-26-07582-f002:**
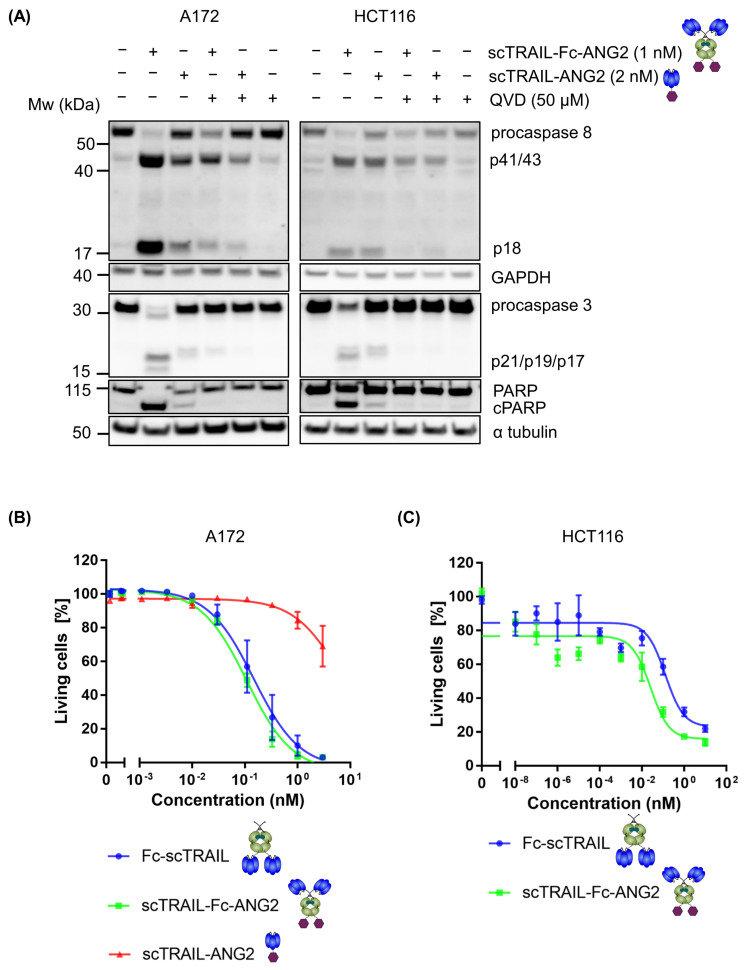
ANG2 fusion does not affect the hexavalent TRAIL potency against cancer cells. (**A**) A172 and HCT116 cells were treated with 1 nM of scTRAIL-Fc-ANG2 or 2 nM of scTRAIL-ANG2 for 6 h with or without 50 µM QVD and blotted for procaspase 8, cleaved caspase 8 (p18), procaspase 3, cleaved caspase 3 (p21/p19/p17), PARP and cleaved PARP (cPARP). GAPDH and α tubulin served as loading controls. Representative western blots from two independent experiments are shown. (**B**) A172 glioblastoma cells were treated with varying concentrations of indicated construct for 24 h and viable cells were determined by Annexin V-PI negativity using flow cytometry. Data are shown as mean ± SEM from three independent experiments. (**C**) Dose-dependent response of HCT116 colon cancer cells to Fc-scTRAIL or scTRAIL-Fc-ANG2 after 16 h stimulation with different concentrations of the constructs. Cell viability was measured by crystal violet assay. Data show results from one representative experiment.

**Figure 3 molecules-26-07582-f003:**
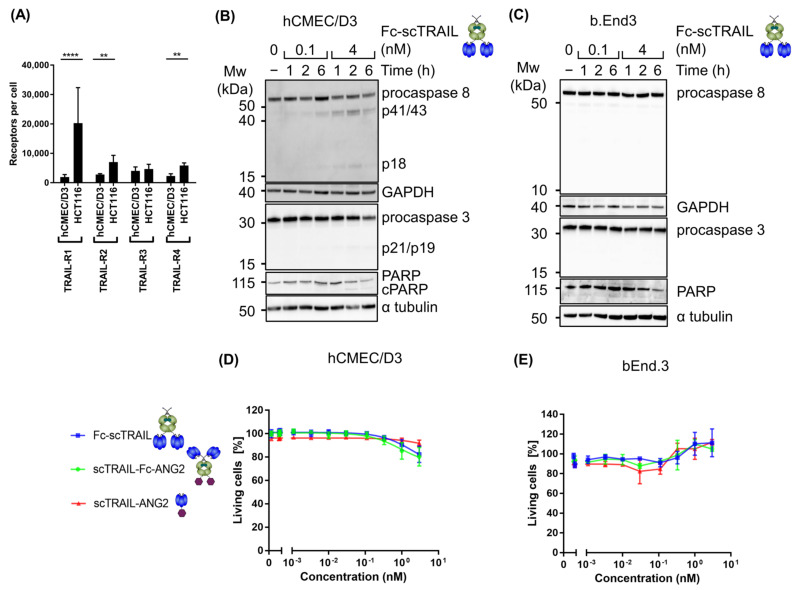
Blood–brain barrier cells are highly resistant to TRAIL treatment. (**A**) Surface expression of TRAIL-R1, TRAIL-R2, TRAIL-R3, TRAIL-R4 receptors in hCMEC/D3 and HCT116 cells were quantified by the QIFIKIT and flow cytometry. The mean ± SD of three independent experiments is shown. Statistical significance was tested by Tukey’s two-way ANOVA on log-transformed receptor values: **** = *p* < 0.0001, ** = *p* < 0.01. (**B**) hCMEC/D3 cells were treated with the indicated concentration of Fc-scTRAIL for 1, 2 or 6 h and then analysed for procaspase 8, cleaved caspase 8 (p18/p10), procaspase 3 and cleaved caspase 3 (p21/p19/p17) by western blotting. Representative images from two independent experiments are shown. (**C**) bEnd.3 cells were treated with the indicated concentration of Fc-scTRAIL for 1, 2 or 6 h and then analysed for procaspase 8, cleaved caspase 8 (p18/p10), procaspase 3 and cleaved caspase 3 (p21/p19/p17) by western blotting. Representative images from two independent experiments are shown. (**D**) hCMEC/D3 cells were treated with indicated construct for 24 h. Viable cells were determined by Annexin V-PI negativity using flow cytometry. Data are shown as mean ± SEM from three independent experiments. (**E**) bEnd.3 cells were treated with indicated construct for 24 h. Viable cells were determined by MTT assay. Data are shown as mean ± range from two independent experiments.

**Figure 4 molecules-26-07582-f004:**
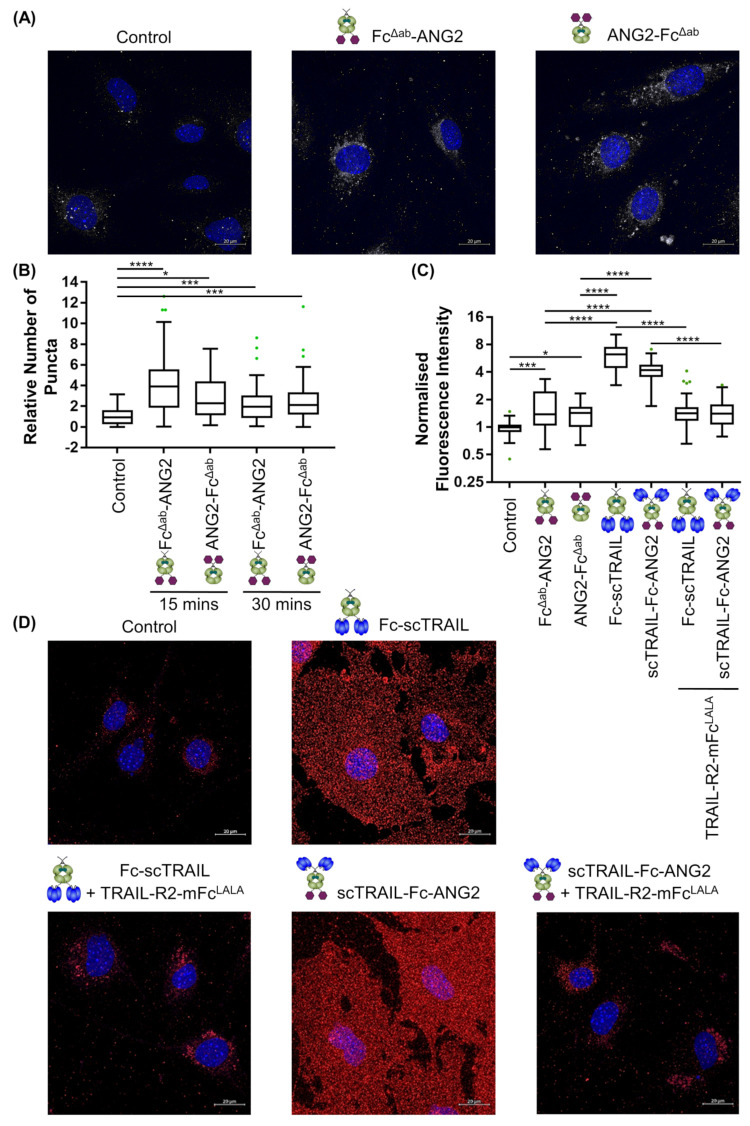
Binding to blood–brain barrier cells is predominantly TRAIL-mediated. (**A**) bEnd.3 cells were incubated with 50 nM of indicated ANG2-control protein for 30 min. Nuclei were stained in blue with Hoechst. White signal indicates the binding of the constructs. Scale bar 20 µm. (**B**) The number of vesicles from A were quantified at 15 min and 30 min and compared between control and ANG2-control protein conditions. Data were plotted as a Tukey boxplot with outliers represented as individual points, minimum of 44 cells per group pooled from three independent experiments. Statistical significance was tested by Tukey’s two-way ANOVA: * = *p* ≤ 0.05; *** = *p* ≤ 0.001 **** = *p* ≤ 0.0001. (**C**) bEnd.3 cells were incubated with 50 nM of indicated construct for 15 min with or without 30 min pre-incubation of 100-fold molar excess of TRAIL-R2-mFc^LALA^. The fluorescence intensity of the cells was quantified and compared between control and construct condition. Data were plotted as a Tukey boxplot with outliers represented as individual points, minimum 43 cells per group pooled from three independent experiments. Statistical significance was tested by non-parametric one-way ANOVA, Kruskal–Wallis test with Dunn’s correction: * = *p* ≤ 0.05; *** = *p* ≤ 0.001 **** = *p* ≤ 0.0001. (**D**) Representative images from C, nuclei were stained in blue with Hoechst. Red signal indicates binding of the constructs. Secondary antibody, anti-Fc-PE was used as a control for non-specific signal. Scale bar 20 µm.

**Figure 5 molecules-26-07582-f005:**
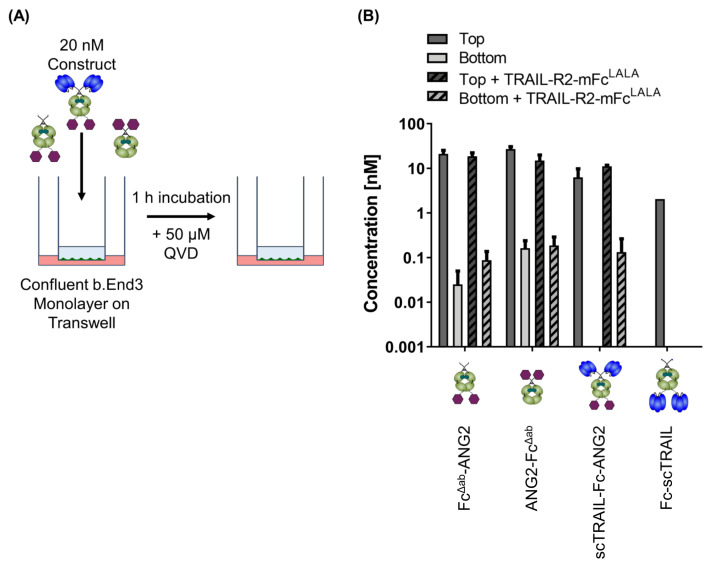
Transwell assay for determination of CNS-transport efficiency (**A**). bEnd.3 cells were grown to a confluent monolayer on a transwell insert and then 20 nM of indicated construct, with or without 30 min pre-incubation of 100-fold molar excess of TRAIL-R2-mFc^LALA^, was added to the apical compartment. (**B**) After 60 min incubation at 37 °C, the samples were taken from the top and bottom compartment of the transwell and the concentration was determined through quantitative sandwich ELISA. Data points are mean + range from two independent experiments.

## Data Availability

All processed data for the study are included within the manuscript and Appendix A. Raw datasets are available from the corresponding author on reasonable request.

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
