# Peer review of "Low-Level Endothelial TRAIL-Receptor Expression Obstructs the CNS-Delivery of Angiopep-2 Functionalised TRAIL-Receptor Agonists for the Treatment of Glioblastoma"

_molecules, 2021, doi:10.3390/molecules26247582_

Round 1

Reviewer 1 Report

This manuscript by Moorthy et al. describes how a new fusion protein, namely sc-TRAIL-fc-ANG2 can retain both its ability to induce apoptosis and to translocate through BBB endothelial cells. However, this translocation ability is conditioned to the presence of TRAIL-R masking entities.

Overall, I think this manuscript should be published provided some points are properly addressed.

1- Given that binding of the constructs to BBB ENDOTHELIAL cells (and not BBB cells) is mainly medited through TRAIL (figure 4), how comes that the trafficking of TRAIL-only constructs was not explored through endothelial cells (figure 5) ? This is an important point to explain, because (i) TRAIL-based constructs were found safe for (= did not induce apoptosis in) endothelial cells; (ii) there was no follow up on what was the fate of endocytosed TRAIL-based constructs in endothelial cells (figure S2) ; (iii) the trafficking rate of ANG-2-based constructs is very low by itself ?

2- Figure 5: filter-only (=no cells) controls shoud be provided, as the provided TEER value  is excessicely low compared to what is generally observed for such set ups. This could be a SI figure but should be present (and discussed).

3- Figure 5 and corresponding discussion section: why did the authors quantify the translocation at 1 h and not at further times ? Considering the binding/translocation kinetics ?

4- Considering the in vitro translocation rate of ANG-2-based constructs (less than 1% from what I could deduce from figure 5, these numbers should be added to help in the discussion), are these ANG2-based fusion proteins really viable for clinical applications ?

5- The fact that the more relevant experiments (figures 4 and 5) were performed on murine cell line and not a human cell line (because of a weaker response to the developped strategy) should be discussed a little bit more. Were there no alternative human cells available ? What about the work scopes ?

Minor considerations:

  • Throughout the manuscript : authors used ENDOTHELIAL bbb cells, therefore they can not conclude for the whole BBB (endothelial cells + pericytes + astrocyte endfeet + ...). Please correct this imprecision by replacing BBB cells by BBB endothelial cells.
  • LRP1 (or Lrp1) should be written the same throughout the manuscript.
  • Section 2.1., line 109: Maldi-tof spectra should be provided in SI.
  • Section 2.2., line 122: please present your cell lines here, instead of further down in the text (lines 130-132)
  • Section 2.2, line 131: please state actual IC50.
  • Figure 4 : this figure should be prepared anew, with bigger pictures so we can actually see that quantification was performed in similar conditions, and a better organization.

Reviewer 2 Report

I found the presented manuscript on the delivery of angiopep-2 functionalised  TRAIL-Receptor agonists interesting and intriguing. In my opinion, the presented results are significant and have consequences for the design of fusion proteins, that are thought to be aimed at specific cells. As the authors demonstrate, there are unexpected traps that we should be aware of.   Nevertheless, I found few issues that should be addressed.
  1. The message of the article (interference of two elements of constructs due to affinity difference) should be stated more clearly and more discussed in the Discussion section.
  2. In lines, 95-97 authors write: "To determine if C-terminal fusion was only a convention or 95 serves functional purposes, we created two separate ANG2-positive control proteins by 96 fusing ANG2 to the C- or N- terminal ... "
    That question is valid and interesting but was not addressed in the result description or Discussion section. Please do so.
  3. In lines, 285-289 authors state: "In this paper, we demonstrate that ANG2-functionalised hexavalent TRAIL receptor agonists retain their potency in killing GBM cells, that BBB cells are resistant to TRAIL and able to transport large TRAIL-fusion proteins across the BBB. However, transport is only possible when the TRAIL binding to BBB cells is blocked."
    I disagree. This message is in a way twisted. Authors have shown that the ANG2-trail construct can't pass BBB. Blocking the TRAIL may solve that problem, but this does not change the fact that the TRAIL-ANG2 constructs did not pass the BBB, by themselves. Please correct.
  4. On figures, the compounds icons are very informative. But could it be bigger?

Reviewer 3 Report

In the study, authors developed a TRAIL and angiopep-2 fusion protein for glioma targeting delivery of TRAIL. The idea is interesting, but experimental design is poor, and I think the conclusion proposed in title is wrong.

  1. Over half of the abstract is about the background. More results should be added in the abstract.
  2. “therapeutic avenues will require combinatorial strategies, such as TRAIL-R masking, to achieve effective CNS- transport”. Results showed the low TRAIL receptor expression on BBB cells interferes with efficient transport. Then, researchers should elevate the TRAIL receptor for improving CNS transport, I don’t think TRAIL-R masking, which further reduces available receptor, will work.
  3. I don’t think the low TRAIL receptor on BBB obstructs CNS delivery of TRAIL-Ang. The TRAIL receptor is a therapeutic target, if the receptor is highly expressed on BBB, the fusion protein would lead serious toxicity to BBB, and the fusion protein could not be used for glioma treatment.
  4. Why did authors think the binding of fusion protein to BBB is predominantly TRAIL-mediated. What’s the function of Angiopep-2?

Reviewer 4 Report

LINE 46 in vivo and line 260 in vitro change by italics

As part of the ethical considerations add information where the project was approved

Round 2

Reviewer 3 Report

Basically, I think the design is not proper. Authors said the BBB endothelial cell has TRAIL receptor and is resistant to TRAIL. My concern is the TRAIL receptor is a therapeutic target. In glioma cells, the TRAIL binds with its receptor to induce apoptosis, so it is hard to understand that in another cell, the binding does not have any toxicity. Additionally, if the BBB penetration was dominated by the binding of TRAIL, why did authors use angiopep-2? Normally, angiopep-2 is a targeting ligand that mediates BBB penetration.

Author Response

The reviewer states that as TRAIL is a therapeutic target that it is difficult to understand that it does not have toxicity in other cells. It should be noted that TRAIL receptors have been shown at both RNA and protein levels to be widely expressed in normal human tissues (https://www.proteinatlas.org/ENSG00000104689-TNFRSF10A;https://www.proteinatlas.org/ENSG00000120889-TNFRSF10B; DANIELS et al., 2005), therefore some TRAIL receptor expression on blood-brain barrier cells was not unexpected. In in vivo models (Gieffers et al., 2013; Hutt et al., 2017) and human clinical trials (https://ascopubs.org/doi/abs/10.1200/JCO.2019.37.15_suppl.3013), TRAIL therapeutics were shown not to cause systemic toxicity and cell death despite this wide expression of TRAIL-receptors. Instead, TRAIL has been shown to induce apoptosis exclusively in cancer cells (Ashkenazi et al., 1999; Walczak et al., 1999). Importantly, expression of TRAIL receptors alone has been shown not to be predictive of TRAIL-sensitivity itself (DANIELS et al., 2005; Phillips et al., 2021). Instead, numerous mechanisms of TRAIL-mediated apoptosis regulation and resistance have been well-established (Zhang & Fang, 2004). Beyond innate cancer cell defence, endogenous TRAIL and its TRAIL receptors also fulfil other functions including in immune system regulation and alternative signalling pathways (Chyuan & Hsu, 2020; Hu, Johnson, & Shu, 1999). Taken together with our data from figure 3, our conclusion that TRAIL receptor is expressed at the BBB but that these cells are resistant to TRAIL-induced apoptosis is in line with current literature on TRAIL-based therapeutics. The reviewer has not provided a rationale as to why this conclusion, based on a rigorous collection of independent studies, is incorrect.
Regarding the point about BBB penetration being dominated by the binding of TRAIL and why angiopep-2 was used. We reiterate that TRAIL itself does not mediate BBB-penetrance; this would be a nonsensical assumption based on the known biology and function of TRAIL-based signalling. We show that TRAIL-mediates binding to BBB endothelial cells (Figure 4C and D), however, our data shows this does not consequent in penetrance. In Figure 5B, constructs that include TRAIL are not detected in the basolateral compartment. Therefore, we concluded that, as expected, TRAIL alone is not able to penetrate across BBB endothelial cells. In contrast, ANG2-only constructs are found in the basolateral compartment, demonstrating that ANG2 mediates BBB-penetrance as expected.
References
Ashkenazi, A., Pai, R. C., Fong, S., Leung, S., Lawrence, D. A., Marsters, S. A., … Schwall, R. H. (1999). Safety and antitumor activity of recombinant soluble Apo2 ligand. The Journal of Clinical Investigation, 104(2), 155–162.https://doi.org/10.1172/JCI6926
Chyuan, I.-T., & Hsu, P.-N. (2020). TRAIL regulates T cell activation and suppresses inflammation in autoimmune diseases. Cellular & Molecular Immunology, 17(12), 1281–1283. https://doi.org/10.1038/s41423-020-0410-2
DANIELS, R. A., TURLEY, H., KIMBERLEY, F. C., LIU, X. S., MONGKOLSAPAYA, J., CH’EN, P., … SCREATON, G. R. (2005). Expression of TRAIL and TRAIL receptors in normal and malignant tissues. Cell Research, 15(6), 430–438. https://doi.org/10.1038/sj.cr.7290311
Gieffers, C., Kluge, M., Merz, C., Sykora, J., Thiemann, M., Schaal, R., … Hill, O. (2013). APG350 Induces Superior Clustering of TRAIL Receptors and Shows Therapeutic Antitumor Efficacy Independent of Cross-Linking via Fcγ Receptors. Molecular Cancer Therapeutics, 12(12), 2735 LP – 2747. https://doi.org/10.1158/1535-7163.MCT-13-0323
Hu, W.-H., Johnson, H., & Shu, H.-B. (1999). Tumor Necrosis Factor-related Apoptosis-inducing Ligand Receptors Signal NF-κB and JNK Activation and Apoptosis through Distinct Pathways. Journal of Biological Chemistry , 274(43), 30603–30610. https://doi.org/10.1074/jbc.274.43.30603
Hutt, M., Marquardt, L., Seifert, O., Siegemund, M., Müller, I., Kulms, D., … Kontermann, R. E. (2017). Superior properties of Fc-comprising scTRAIL fusion proteins. Molecular Cancer Therapeutics.
Phillips, D. C., Buchanan, F. G., Cheng, D., Solomon, L. R., Xiao, Y., Xue, J., … Morgan-Lappe, S. E. (2021). Hexavalent TRAIL Fusion Protein Eftozanermin Alfa Optimally Clusters Apoptosis-Inducing TRAIL Receptors to Induce On-Target Antitumor Activity in Solid Tumors. Cancer Research, 81(12), 3402 LP – 3414. https://doi.org/10.1158/0008-5472.CAN-20-2178
Walczak, H., Miller, R. E., Ariail, K., Gliniak, B., Griffith, T. S., Kubin, M., … Lynch, D. H. (1999). Tumoricidal activity of tumor necrosis factor–related apoptosis–inducing ligand in vivo. Nature Medicine, 5, 157. Retrieved fromhttps://doi.org/10.1038/5517
Zhang, L., & Fang, B. (2004). Mechanisms of resistance to TRAIL-induced apoptosis in cancer. Cancer Gene Therapy, 12, 228. Retrieved from https://doi.org/10.1038/sj.cgt.7700792